# *Candida glabrata* Antifungal Resistance and Virulence Factors, a Perfect Pathogenic Combination

**DOI:** 10.3390/pharmaceutics13101529

**Published:** 2021-09-22

**Authors:** María Guadalupe Frías-De-León, Rigoberto Hernández-Castro, Esther Conde-Cuevas, Itzel H. García-Coronel, Víctor Alfonso Vázquez-Aceituno, Marvin A. Soriano-Ursúa, Eunice D. Farfán-García, Esther Ocharán-Hernández, Carmen Rodríguez-Cerdeira, Roberto Arenas, Maura Robledo-Cayetano, Tito Ramírez-Lozada, Patricia Meza-Meneses, Rodolfo Pinto-Almazán, Erick Martínez-Herrera

**Affiliations:** 1Unidad de Investigación, Hospital Regional de Alta Especialidad de Ixtapaluca, Ixtapaluca 56530, Mexico; magpefrias@gmail.com (M.G.F.-D.-L.); mrobledoc@hotmail.com (M.R.-C.); 2Departamento de Ecología de Agentes Patógenos, Hospital General “Dr. Manuel Gea González”, Ciudad de México 14080, Mexico; rigo37@gmail.com (R.H.-C.); leonovictor84@gmail.com (V.A.V.-A.); 3Maestría en Ciencias de la Salud, Escuela Superior de Medicina, Instituto Politécnico Nacional, Plan de San Luis y Díaz Mirón, Ciudad de México 11340, Mexico; condeesther999@gmail.com (E.C.-C.); itzelhaydeg@gmail.com (I.H.G.-C.); patricia_meza@ymail.com (P.M.-M.); 4Sección de Estudios de Posgrado e Investigación, Escuela Superior de Medicina, Instituto Politécnico Nacional, Plan de San Luis y Díaz Mirón, Ciudad de México 11340, Mexico; msoriano@ipn.mx (M.A.S.-U.); efarfang@ipn.mx (E.D.F.-G.); estherocharan@hotmail.com (E.O.-H.); 5Efficiency, Quality, and Costs in Health Services Research Group (EFISALUD), Galicia Sur Health Research Institute (IIS Galicia Sur), SERGAS-UVIGO, 36213 Vigo, Spain; carmencerdeira33@gmail.com (C.R.-C.); rarenas98@gmail.com (R.A.); 6Dermatology Department, Hospital Vithas Ntra. Sra. de Fátima and University of Vigo, 36206 Vigo, Spain; 7Campus Universitario, University of Vigo, 36310 Vigo, Spain; 8Sección de Micología, Hospital General “Dr. Manuel Gea González”, Tlalpan, Ciudad de México 14080, Mexico; 9Servicio de Ginecología y Obstetricia, Hospital Regional de Alta Especialidad de Ixtapaluca, Ixtapaluca 56530, Mexico; titolozada@yahoo.com.mx; 10Servicio de Infectología, Hospital Regional de Alta Especialidad de Ixtapaluca, Ixtapaluca 56530, Mexico

**Keywords:** *Candida glabrata*, antifungal resistance, variations in drug resistance, virulence factors, adherence mechanisms, enzymatic activity, formation of biofilms

## Abstract

In recent years, a progressive increase in the incidence of invasive fungal infections (IFIs) caused by *Candida glabrata* has been observed. The objective of this literature review was to study the epidemiology, drug resistance, and virulence factors associated with the *C. glabrata* complex. For this purpose, a systematic review (January 2001–February 2021) was conducted on the PubMed, Scielo, and Cochrane search engines with the following terms: “*C. glabrata* complex (*C. glabrata sensu stricto*, *C. nivariensis*, *C. bracarensis*)” associated with “pathogenicity” or “epidemiology” or “antibiotics resistance” or “virulence factors” with language restrictions of English and Spanish. One hundred and ninety-nine articles were found during the search. Various mechanisms of drug resistance to azoles, polyenes, and echinocandins were found for the *C. glabrata* complex, depending on the geographical region. Among the mechanisms found are the overexpression of drug transporters, gene mutations that alter thermotolerance, the generation of hypervirulence due to increased adhesion factors, and modifications in vital enzymes that produce cell wall proteins that prevent the activity of drugs designed for its inhibition. In addition, it was observed that the *C. glabrata* complex has virulence factors such as the production of proteases, phospholipases, and hemolysins, and the formation of biofilms that allows the complex to evade the host immune response and generate fungal resistance. Because of this, the *C. glabrata* complex possesses a perfect pathogenetic combination for the invasion of the immunocompromised host.

## 1. Introduction

Since the late 20th century, a progressive increase in the incidence of invasive fungal infections (IFIs) has been observed, with those caused by species of the genus *Candida* being more frequent (43–75%) [1]. *Candida albicans* is the most frequent causative agent of IFIs [2,3,4,5,6,7,8,9,10,11,12,13,14,15,16,17,18,19,20,21,22,23,24,25,26]; however, *C. glabrata* is increasingly being isolated in cases of invasive candidiasis [5,6,9,11,12,18,20,23,25,26,27,28,29,30,31,32,33,34,35,36] and is associated with increased mortality in patients [4,5]. This increase in the incidence of infections by *C. glabrata* and other non-*albicans* species is indirectly attributed to the development of diverse technologies and current medical treatments, such as organ transplantation, the use of medical devices of different materials such as urinary and vascular catheters, intrauterine devices, pacemakers, prosthetic valves, as well as broad-spectrum antibiotic therapies [2,3,28,31,32,37,38,39,40,41,42,43,44,45]. *C. glabrata* is a non-hyphae-producing haploid yeast described in 1917 by Harry Warren Anderson as part of the intestinal biota called *Cryptococcus glabratus* [46,47]. However, it was not until 1995 that Kevin C. Hazen recognized *C. glabrata* as an emerging pathogenic yeast commonly found in patients with diabetes mellitus, solid tumors, malnutrition, in neonates, and sometimes in patients with hematologic neoplasms [48]. *C. glabrata* has been considered a commensal and opportunistic fungus of the digestive system, which, like *C. albicans,* can become lethal in immunocompromised patients [40,49,50].

The ecological niche of *C. glabrata* is highly specialized but can adapt to different microenvironments to proliferate efficiently within the host [41,49]. Despite its adaptation to humans, *C. glabrata* has been isolated from abiotic surfaces in hospital settings and from the hands of health personnel [50,51,52,53], as well as outside clinical settings (fermentation of coffee beans and feces of various bird species) [50,54].

Molecular studies have shown that *C. glabrata* is more closely related to *Saccharomyces cerevisiae* than *C. albicans*. Thus, in 2003, Krutzman (2003) classified *C. glabrata* within the *Nakaseomyces* clade as it was not initially considered a human pathogen. Between 2005 and 2006, the molecular analysis of *C. glabrata* clinical isolates revealed the existence of two species, *Candida nivariensis* and *Candida bracarensis*, which are indistinguishable from *C. glabrata* at the phenotypic level but genetically distinct and with great potential to cause opportunistic infections. These species composed the *C. glabrata* complex [55,56,57,58,59,60,61]. However, due to the difficulty differentiating them from *C. glabrata sensu stricto*, the clinical significance and actual prevalence of *C. nivariensis* and *C. bracarensis* have been underestimated. The latter affects their treatment, as they exhibit greater resistance to azoles and echinocandins [57,58,61,62,63,64]. It should be noted that the increasing use of azole antifungals for the treatment of superficial and systemic infections by *Candida glabrata* has led to the selection and emergence of resistant isolates, as well as increased infections by other non-*albicans* species [57]. Besides, it is considered that *C. glabrata sensu stricto* is the most virulent species, followed by *C. nivariensis* and *C. bracarensis* [62,63].

The different *Candida* species have virulence factors that contribute to their pathogenicity, especially in immunocompromised patients [9,65]. *C. glabrata* can secrete hydrolytic and proteolytic enzymes that allow its adherence to host cells and invasive medical devices [39,66,67]. Likewise, it is capable of forming biofilms that provide antifungal resistance and also exhibit great advantageous genomic plasticity. Therefore, it is considered as an aggressive yeast for humans [39,66,67].

Information about the various virulence factors used by the *C. glabrata* complex species is scarce [57,58,63]. However, it is important to be acquainted with the virulence factors shown by the *C. glabrata* complex due to the progressive increase of infections caused by these fungi and their high rate of antifungal resistance [2,24,29,51,68,69,70,71].

Therefore, the objective of this work is to conduct a systematic review of the virulence factors attributed to the *C. glabrata* complex, and the current state of antifungal resistance of the species that form this complex.

## 2. Materials and Methods

During May 2021, an advanced search was conducted in the engines PubMed, Scielo, and Cochrane, for the period comprised between 2001 and 2021, with the terms: “*Candida glabrata* complex (*C. glabrata sensu stricto, C. nivariensis*, *C. bracarensis*)” associated with “pathogenicity” or “epidemiology” or “antibiotics resistance” or “virulence factors”. The total number of articles found was 199. The review was performed based on the preferred reporting items for systematic reviews and meta-analyses (PRISMA) (Figure 1).

## 3. Immunological Regulation of the Host

During their evolution, *Candida* spp. yeasts have managed to develop strategies to grow inside human hosts, which have a wide variety of niches for growth. The various *Candida* species require rapid and coordinated changes in their metabolism and physiology to avoid or escape the immune surveillance and adapt to an adverse or constantly changing microenvironment [72,73]. Such strategies allow the fungus to survive in environments with nutrient limitations, antimicrobial peptide production, pH fluctuations, oxygen deprivation, temperature changes or with oxidative, nitrosamine, and cationic stress [72,74].

The fungal cell wall is responsible for mediating the initial steps of the host–fungus interaction for their mutual recognition and activation of the signal transduction through the host receptors [73,75]. The accessibility to the β-glucan and chitins present in the fungal cell wall is crucial for the recognition and activation of the host’s immune system [73].

As a mechanism of immune evasion, the species of the genus *Candida* have successfully managed to mask their cell membrane’s pathogen-associated molecular patterns (PAMP) secreting specific proteases aimed at the opsonization of the complement, to avoid being recognized by the multiple pattern recognition receptors (PRRs) present in the cells of the innate immune system (dendritic cells, macrophages, monocytes, and neutrophils) that are responsible for inducing phagocytosis and the production of cytokines and proinflammatory chemokines [72,73].

When the host’s immune system recognizes the components of the fungal cell wall, especially the β-glucan or chitin, the phagocytosis and production of proinflammatory cytokines are triggered [73]. The phagocytes will attempt to kill pathogens by producing toxic reactive oxygen species (ROS) and reactive nitrogen species (RNS) as an antimicrobial defense mechanism [72,73,76]. When the ROS interact with nitric oxide (NO), they produce toxins that irreversibly damage the pathogen, thus fighting the invasion [72].

Seider et al. noted that β-glucan and chitin are vital to produce tumor necrosis factor-alpha (TNF-α) in *C. glabrata*. The unmasking of both components in the fungus causes an inflammatory response altered by monocyte-derived macrophages (MDMs) by induction of the TNF-α. The higher the chitin content, the greater the production of TNF-alpha [73]. Likewise, the increased exposure to β-glucan and chitin was also associated with increased ROS production [73].

The genus *Candida* species can induce their capture through phagocytes in endothelial and epithelial cells, using them as a “safe home”, preventing the maturation of the phagolysosome and its destruction [72,77]. It has been reported that *C. glabrata* can survive within the phagosome due to its ability to generate strong stress responses against ROS, neutralizing the phagocytic environment and thus escaping phagocytosis [72,77,78,79,80]. Such a phenomenon requires the activation of different genes in the pathogen (*Skn7p*, *Yap1p*, *Msn2p*, and *Msn4p*) [76,81,82] that encode proteins (Transcription factors Skn7, CgYap1, MSN2p, MSN4p) involved in detoxification and repair such as catalases, superoxide dismutases, glutathione peroxidases, and thioredoxins [76,80,81,83]. Both in *C. albicans* and *C. glabrata* these stress pathways are mainly regulated by the stress-activated protein kinase (SAPK) Hog1 [73,84,85], the Cap1 transcription factor, and the DNA damage checkpoint kinase Rad53, which, together with the CTA4 transcription factor, play an important role in triggering the response to osmotic, oxidative, and nitrosative stress [73,85]. It has been observed that deleting these genes results in virulence attenuation, alters stress tolerance, and therefore decreases phagocytic survival [72]. It is also known that, according to that mentioned above, *C. glabrata* has important antioxidant systems, so the ROS play a minor role in destroying this pathogen [73].

Certain genes like *CgVPS15* and *CgVPS3*4 (orthologs of *Vps15* and *Vps34* in *S. cerevisiae*) have also been identified to be relevant for regulating the phosphatidylinositol 3’-kinase (PI3K) signaling pathway. It is known that in *C. glabrata*, the PI3K signaling is essential for the microorganism’s survival inside the host’s macrophages. In a mouse model of systemic infection, the deletion of *CgVPS15* and *CgVPS34* caused altered protein trafficking in *C. glabrata* and high susceptibility to stress, as well as hyperadherence to epithelial cells, which led to the yeast not being able to survive inside the macrophages. The study concluded that both genes are needed to modulate the phagolysosome acidification and survival in macrophages, and that fungal PI3K is critical for the pathogenesis and virulence of *C. glabrata* [86].

Micronutrient limitation is also an effective way to control the proliferation of pathogens [72,77,81]. This is especially true for micronutrients such as iron and zinc, which are important for the development of intracellular pathogens such as *C. glabrata* [87]. Upon such restriction, *C. glabrata* has managed to develop strategies to obtain micronutrients that allow proliferation, survival, and even greater resistance to phagocytosis [72,73]. *C. glabrata* can acquire free iron and iron from iron-binding proteins such as hemoglobin, ferritin, and transferrin. The siderophore-iron transporter Sit1 is responsible for mediating the iron acquisition, giving the microorganism the ability to survive phagocytosis and replicate within the host’s macrophages due to the use of intracellular iron deposits [72]. Likewise, it can escape immunological surveillance by kidnapping zinc in vacuoles to regulate Zn homeostasis [72,77]. Furthermore, it has been observed that the deletion of ZRC1 (involved in the regulation of Zn kidnapping in *S. cerevisiae*) in *C. glabrata* results in a survival defect and is crucial for immune evasion [77].

It has also been observed that interferons such as IFN-I play a crucial role in favoring *C. glabrata* persistence in organs such as the brain, liver, and spleen in murine models, causing dysregulation of the cellular iron homeostasis, thus facilitating its acquisition by the pathogen [88]. IFN-I has an important role in the regulation of Zn homeostasis as it reduces plasmatic concentrations of this micronutrient by inducing the expression of hepatic metallothionein, which captures Zn for cytoplasmic transport during the intracellular mobilization of Zn. Studies show that IFN-I suppresses metallothionein gene expression after infection with *C. glabrata* in in vitro and in vivo models [77].

Rasheed et al. suggest in their study that CgYapsins (encoded by the CgYPS1-111 genes) inhibit IL-1β production in macrophages so that the microorganism can proliferate and spread. The increased IL-1β output is part of the host’s immune system’s response to infection [89]. The study conducted in a murine model observed that a mutant strain of *C. glabrata* lacking the coding genes for CgYapsins showed less virulence and died inside the macrophages. Therefore, the authors suggest that given the absence of CgYapsins, the IL-1β-dependent inflammatory response is not inhibited inside the macrophages. Consequently, the microorganism dies as it lacks the aspartyl proteases that contribute to its survival [89].

The dependence between the IL-1β output and CgYapsins occurs because yapsins inhibit IL-1β production in macrophagues. Thus, the pathogen survives the host’s defense mechanism, proliferating and subsequently spreading. Conversely, in the absence of yapsins, there is no inhibition of IL-1β, and the pathogen dies inside the macrophages [89].

## 4. Antifungal Resistance of the *C. glabrata* Complex

*C. glabrata* is characterized by the exhibition of a high resistance rate to the different antifungal therapies currently available [11,52,53,70,74,87,90,91]. Low susceptibility to azoles, polyenes, and echinocandins used in the treatment of invasive or serious infections caused by the *C. glabrata* complex has been associated with a higher mortality rate [29,90,92]. In addition, recent reports indicate a high resistance of *C. glabrata* to medicines such as caspofungin and micafungin, which is alarming [11,52,87]. Because of this, the Infectious Diseases Society of America guidelines initially recommend treatment with echinocandins in patients with candidemia and risk factors for developing infections caused by microorganisms of the *C. glabrata* complex [93,94].

### 4.1. Resistance to Azoles

As mentioned above, one of the known virulence factors of the *C. glabrata* complex is its intrinsic low susceptibility to azoles, especially fluconazole [7,51,53,67,95,96,97]. In general, this is because azoles are the first prophylactic choice against fungal infections due to their low cost, and the second choice for invasive infections produced by different *Candida* species, generating cross-resistance to the other azoles [96,98,99,100].

On the other hand, it has been observed that in different world regions, the pharmacological susceptibility to azoles presents variations as well as the proportion of cases between the types of candidiasis (Table 1) [3,64,101,102,103,104].

### 4.2. Susceptibility Variations per Continent

The study conducted by Tsega et al., in 2019, in Ethiopia, Africa, reported 17 *C. glabrata* strains isolated from pregnant women. In this study, all strains were sensitive to fluconazole. However, 2 exhibited resistances to clotrimazole, 13 to itraconazole, and 7 to ketoconazole [14]. In Cameroon, Ngouana et al., in 2019, analyzed *C. glabrata* strains obtained from mucous membranes of HIV-infected patients. Thirty-three *C. glabrata* strains were isolated with one being resistant to fluconazole (CMI = 0.25–64 μg/mL) [104]. In Ghana, Waikhom et al., in 2020, analyzed clinical isolates obtained from 176 pregnant patients. Only 54 patients were diagnosed with *Candida* infection with positive isolates (44 symptomatic and ten asymptomatic). *C. glabrata* was isolated in 25 symptomatic women and six asymptomatic women, being the most common isolation with 57.4%. Six *C. glabrata* isolates were susceptible to fluconazole (19.4%), 13 were susceptible dose-dependently (41.9%), and 12 were resistant (38.7%). No *C. glabrata* isolate was susceptible to nystatin, 27 were susceptible dose-dependently (87.1%), and 4 were resistant (12.9%). Seventeen strains were susceptible to voriconazole (54.8%), three susceptible dose-dependently (9.7%), and eleven strains were resistant (35.5%) [105].

In the American continent, Fuller et al. reported the isolation of 392 *C. glabrata* strains in Canada, of which 87.8% were dose-dependently susceptible to fluconazole with a minimum inhibitory concentration (MIC) of ≤4 mg/mL [37]. In the U.S.A., it has been observed that infections produced by *C. albicans* have decreased, while those caused by *C. glabrata* have increased up to 25% from the *Candida* non-albicans infections [100]. In South America, there are reports from Chile and Brazil regarding the *C. glabrata* complex. In Chile, Santolaya et al. reported 37 cases of candidemia generated by *C. glabrata*; from which 6.6% were resistant to fluconazole, 20% to itraconazole, 10% to micafungin (two strains with MIC values = 0.25μg/mL and one strain with MIC = 0.5μg/mL), and 6.6% had elevated ECV values for amphotericin B [16]. Similarly, Savastano et al. studied 38 *C. glabrata* strains obtained from different surfaces in a hospital environment in Brazil and performed susceptibility tests on 8 strains. In all cases, 100% were sensitive to amphotericin B and nystatin, 50% were resistant to fluconazole, and 12.5% were resistant to miconazole [51].

In Asia, many of the existing studies on antifungal resistance to antibiotics shown by the *C. glabrata* complex have been conducted. In China, Li et al. conducted a retrospective case-control study in which six strains from the *C. glabrata* complex were tested for antifungal susceptibility, observing the following values: Flucytosine (≤4 μg/mL), amphotericin B (≤0.5 μg/mL), fluconazole (4–8 μg /mL), itraconazole (0.125–0.5 μg/mL), and voriconazole (≤0.06–0.5) [103]. Subsequently, the same working group reported that cancer patients with infections caused by dose-dependently susceptible strains of the *C. glabrata* complex that were given initial monotherapy with azoles were linked to worse outcomes [104]. Likewise, Zeng et al. isolated 73 *C. glabrata* strains from which 11% were resistant to flucytosine (MIC ≤ 0.25 to >8 μg/mL), 6.8% to voriconazole (MIC ≤ 0.062 to >8 μg/mL), 6.8% to fluconazole (MIC ≤ 0.25 to > 8 μg/mL), and 100% of the isolated strains were resistant to amphotericin B [20].

In India, there are studies with different results regarding the antifungal sensitivity exhibited by the *C. glabrata* complex. In a study conducted by Jain et al., 100% of the 21 *C. glabrata* strains analyzed showed MIC values within the ranges of antifungal sensitivity [21]. On the other hand, Pote et al. isolated 22 *C. glabrata* strains from clinical samples obtained from three hospitals that showed resistance to clotrimazole, fluconazole, itraconazole, ketoconazole, and nystatin [70]. In Nepal, Subramanya et al. analyzed the antifungal susceptibility of nine *C. glabrata* isolated strains. They found that all strains were susceptible to caspofungin (MIC ≤ 4 μg/mL) and had intermediate sensitivity to amphotericin B (MIC = 8–16 μg/mL). As for fluconazole, only six strains were susceptible (MIC ≤ 4 μg/mL), one showed intermediate sensitivity (MIC = 8–16 μg/mL), and two were resistant (MIC ≥ 32 μg/mL). Regarding voriconazole, eight strains had intermediate sensitivity and one was resistant to this antifungal [95].

Alobaid et al. isolated 11 *C. glabrata* strains from a second-level hospital in Kuwait that showed variable antifungal susceptibility to fluconazole. The authors reported that 64% of these strains were resistant (MIC = 64 μg/mL) and 36% were dose-dependently susceptible (MIC = 32 μg/mL) [107]. Conversely, Al-Baqsami et al. found that 48% (36 strains) of the 75 *C. glabrata sensu stricto* strains isolated in Kuwait were resistant (MIC > 32 μg/mL) to fluconazole and 52% (39 strains) were dose-dependently susceptible (MIC ≤ 32 μg/mL). In addition, five strains presented resistance to micafungin (MIC > 0.125 μg/mL) (three were resistant and two were dose-dependently susceptible), four showed resistance to caspofungin (MIC of ≥0.5 μg/mL), and five strains were resistant to amphotericin B (MIC > 1 μg/mL) [108].

Gülmez et al. studied several cases of fungemia in Turkey, isolating 12 *C. glabrata* strains resistant to fluconazole [109]. Moreover, Kaan et al. identified 83 *C. glabrata* strains, from which 45.8% were resistant to itraconazole, 9.2% to fluconazole, and 43.4% to voriconazole [71].

Similarly, in Europe, Marin et al. reported in Spain that all isolates from this complex were dose-dependently susceptible to fluconazole [101]. Likewise, Ryan et al. isolated 21 *C. glabrata* strains in Ireland, of which 37% showed resistance to fluconazole and 14% to amphotericin B [5]. Sikora et al. isolated 445 *C. glabrata sensu lato* strains in Poland, from which 24 were identified as *C. nivariensis*. It was observed that all strains had an intermediate sensitivity to fluconazole (MIC = 0.25–256 mg/L), 41% were resistant to itraconazole (MIC = 1.5–32 m/L), 50% were resistant to posaconazole (MIC = 1.5–32 mg/L), and 83% were susceptible to voriconazole (MIC = 0.008–2.0 mg/L). All strains were susceptible to echinocandins and amphotericin B [57]. In Greece, from 2009 to 2018, Siopi et al. reported multidrug resistance in C. glabrata as a major concern; however, no specific mechanism was reported [24]. Furthermore, from a multihospital study, Aldejohann et al. reported resistance in the *C. glabrata* complex to several drugs including echinocandins, probably related to gene regulation; specifically, those linked to glucan synthase expression [110]. Similarly, Coste et al. observed resistance to echinocandins linked to the FKS-genes mutation in Switzerland [111]. However, other authors, such as Fraser et al., suggest that resistance is a rare phenomenon in some countries like the United Kingdom [112]. A multicenter study showed the relevance of *C. glabrata* in yeast infections with mixed agents and suggested a potentiation of resistance in those cases [114]. In Jerusalem, Israel et al. examined *C. glabrata* strains that exhibited a susceptible-dose-dependent pattern to fluconazole with MIC values ≤ 32 μg/mL [11].

An epidemiological study conducted in Australia by Boan et al. reported that 22.8% (8/35) of the *C. glabrata* complex isolated strains were resistant to fluconazole and 17.1% (6/35) were not sensitive to caspofungin. It was also reported that the prevalence of this complex has been increasing [3].

### 4.3. Drug Resistance Fluctuations Caused by the Type of Candida and Genetic Variations

The mechanisms of antifungal resistance in the *C. glabrata* complex are still being thoroughly studied (Table 2) [116]. As mentioned above, the *C. glabrata* complex has shown drug resistance to the azoles in several cases, which act by inhibiting the 14-α lanosterol demethylase that is encoded by the *ERG11* gene. The *ERG11* gene is known to participate in ergosterol biosynthesis [97,117]. Interestingly, Hull et al. 2012 found that the isolate of *C. glabrata* (CG156) has an *ERG11* mutation that induces a loss of function associated with cross-resistance to azoles and polyenes. This isolate exchanges ergosterol from the membrane for other sterols such as lanosterol and fecosterol, among others [115].

Various reports have found that antifungal-resistant *C. glabrata* strains have mutations associated with the Pdr1 transcription factor and the overexpression of the ABC-type efflux pumps (ATP-Binding Cassette), mainly *CDR1* and *CDR2* (*Candida* Drug Resistance) [8,118,119,120]. Such pumps can translocate small molecules to the outside of the cell and are regulated by the TAC1 transcription factor [121,122].

Several mutations have been identified in the PDR1 zinc-cluster-containing transcription factor in the *C. glabrata* complex that favor the overexpression of the *CDR1*, *CDR2*, *SNQ2,* and *PDH1* transporters, which are known to carry multiple drugs [117,118,119,120,123]. For example, Hou et al. observed that 14 *C. glabrata* isolated strains with polymorphisms in *PDR1* showed increased resistance to fluconazole (MIC ≥ 64 μg/mL) [124]. On the other hand, Culakova et al. noted that the deletion of *PDR1* decreases cell surface hydrophobicity during biofilm formation, which increases the susceptibility of these mutant strains to different azoles such as fluconazole, bifonazole, itraconazole, ketoconazole, clotrimazole, and miconazole [125]. Likewise, the activation of azole transporters *CDR1* and *CDR2,* alone or in combination, has been associated with antifungal resistance in *C. glabrata* [8]. Farahyar et al. reported that *C. glabrata* strains have drug-resistant *Candida* genes (*CgCDR) CgCDR1* and *CgCDR2*, as well as Fatty Acid Activator 1 (*FAA1*), which are positively regulated twice as much in resistant strains. These results demonstrated that the overexpression of these three genes is associated with azole resistance by modifying the biological transport pathways of hydrophobic compounds and the lipid metabolism in *C. glabrata* [126]. Szweda et al. demonstrated through real-time PCR studies that 13 of 15 azole-resistant strains displayed upregulation of the CDR1 gene encoding the efflux pump. Conversely, no upregulation of the CDR2 expression or ERG11 gene was observed [113]. In addition, the mitochondrial dysfunction associated with the formation of "small mutants", deficient in mitochondrial DNA, positively regulates the ABC transporter genes, increasing resistance to these drugs [64,120,124].

Another factor associated with drug resistance in *C. glabrata* is *ADA2* [64]. *ADA* is a component that serves as a transcription adapter of the Spt-Ada-Gcn5 acetyltransferase complex (SAGA complex), previously found in *C. albicans* and determined to be necessary for tolerance and virulence of antifungal drugs. Shi et al. and Yu et al. observed various roles that *ADA2* has in cellular functions such as growth, cell wall integrity, antifungal tolerance, and suppression of virulence in the *C. glabrata* complex [64,127]. Yu et al. reported that *ADA2* is involved in thermotolerance, finding that mutations on *C. glabrata ADA2* (*CgADA2*) generate severe defects in the growth of strains at 40 °C, and intermediate defects at 37 °C and 25 °C. It also increases the susceptibility of *C. glabrata* towards azoles, echinocandins, and polyenes. However, the authors also reported that the deletion of CgADA2 resulted in hypervirulence of the strains in an in vivo murine model, possibly due to the positive regulation of adherence factors in strains [127].

Since most *C. glabrata* strains are resistant to azoles, a therapeutic measure that has shown some effectiveness is echinocandins. Drugs such as anidulafungin, caspofungin, and micafungin inhibit the glucan synthase enzyme [128]. These drugs inhibit the formation of β-1,3-D glucan by non-competitively binding to the *Fks1p* and *Fks2p* subunits of the β-1,3 glucan synthase. As the β-1,3-D glucan is an integral part of the structure and function of the fungal cell wall, the inhibition of its formation generates high permeability of the cell wall and thus cell lysis [61,108,111,129]. However, increased drug resistance to these drugs has been seen in the *C. glabrata* complex due to previous exposure to these antifungals [111,130]. Resistance to these drugs comes from specific mutations that lead to amino acid substitutions in two different regions of these genes (hotspots 1 and 2 or (*HS1* and *HS2*)), altering the conformation of the *Fks1p* and *Fks2p* subunits, thus lowering affinity to echinocandins [108]. Pan-resistance to these antifungals in *C. glabrata* has a prevalence range of 2–13% [111,131]. In Switzerland, Coste et al. identified five candidemia cases by *C. glabrata* previously exposed to echinocandins and resistant to this drug. After a molecular study, the authors found that drug resistance in three strains was associated with mutation *S663P* in *FKS2*, mutation *S629P* in *FKS1* in one strain, and *F659* in *FKS2* in another one [111]. However, the presence of *FKS* gene mutations in *C. glabrata* isolates is not always associated with phenotypic resistance in vitro [132]. In this regard, in 2012, Katiyar et al. demonstrated that even when *Fks1* and *Fks2* have the same functionality, the redundancy of *Fks1-Fks2* attenuates the resistance rate and the impact of mutations that confer resistance to echinocandins [133].

Another important gene in the generation of antifungal resistance is the *ERG6* as it intervenes in the integrity of the cell wall and the pharmacological tolerance in these yeasts [64]. Vandeputte et al. reported that a *C. glabrata* isolate with a missense mutation in the *ERG6* gene had a lower ergosterol content associated with its biosynthesis pathway, causing cell wall modifications and increased susceptibility to drugs acting on the cell wall [134].

Vacuolar proton-translocating ATPases (V-ATPases) are present in the vacuolar membranes of fungi. They regulate many cellular processes and keep ionic homeostasis by maintaining acidic pH inside the fungal cell. Minematsu et al. conducted a study where they observed that when removing the *VPH2* gene and interrupting the V-ATPase function, the response of *C. glabrata* is altered causing decreased virulence and homeostasis alteration of the vacuolar pH. Therefore, it is assumed that *VPH2* deletion may increase susceptibility to antifungals [135]. Accordingly, Roetzer et al. demonstrated in 2008 that the protein *VPH2*, required for vacuolar H^+^-ATPase function, was widely induced under different oxidative stress conditions in *C. glabrata strains*. These studies allow us to understand that, when subjected to certain conditions, the *C. glabrata* strains can induce the expression of this protein to maintain the internal pH of the cell and preserve its virulence [136].

### 4.4. Drug Resistance Variations in C. Glabrata Complex Species

Reports indicate that in the same way there are differences in the virulence factors among the *C. glabrata* complex species, there are also differences in their antifungal resistance. For example, reports show that *C. glabrata sensu stricto* is more susceptible to fluconazole, itraconazole, and voriconazole than *C. nivariensis* [62,63,92].

López et al. and Fujita et al. reported catheter-associated candidemia resistant to empirical treatment with fluconazole caused by *C. nivariensis*. When conducting susceptibility tests to various antifungals in blood cultures, it was observed that, in these cases, treatments with echinocandins (caspofungin and micafungin) and flucytosine were the most appropriate because of their sensitivity [118,137].

According to Shi et al., the *C. nivariensis* strains presented a higher MIC than *C. albicans*. They also showed increased expression of virulent and resistant genes such as *YPS1*, *AWP3*, *EPA1*, *ERG11*, *CDR1,* and *CDR2* than *C. glabrata sensu stricto* isolates [64]. In addition, in patients with vulvovaginal candidiasis caused by *C. nivariensis*, a low cure rate was observed using conventional antifungals [64].

Accordingly, in the study conducted by Shi et al., it was observed that the mRNA expression of *ERG11*, *CDR1,* and *CDR2* was higher in isolates of *C. nivariensis* strains than *C. glabrata sensu stricto*. These results are linked to *C. nivariensis* strains’ drug resistance [64]. However, Arastehfar et al. showed that the *C. nivariensis* strains obtained from clinical isolates were sensitive to azoles, polyenes, and echinocandins [106].

In turn, Kaur et al. and Sikora et al. showed that the *YPS* gene’s deletion or mutations cause decreased virulence in *C. glabrata* [138,139]. Currently, there is no evidence on the prevalence of *YPS* in *C. nivariensis* and little is known about its antifungal susceptibility [140].

Regarding *C. bracarensis*, Małek et al. performed in vitro susceptibility tests of different antifungals in isolates obtained by PCR from a group of 353 strains of the *C. glabrata* complex, evidencing drug resistance to various azoles (fluconazole, itraconazole, and posaconazole), amphotericin B, and flucytosine. They also reported sensitivity to echinocandins (anidulafungin and caspofungin) [58].

## 5. Virulence Factors

Virulence factors are important for the pathogenicity of *Candida* spp., as they allow colonization, adhesion, invasion, and dissemination in tissues. They also help evade the host’s defenses to cause infection [63,141,142,143].

Enzymatic activity is considered an important virulence factor in *Candida* spp. [144]. However, this type of yeast usually expresses virulence factors in different degrees, according to the species. Moreover, as mentioned previously, its expression may depend on the isolate geographical origin, type of infection, site of infection, and the host’s reaction [145].

Among the virulence factors used by the *C. glabrata* complex are biofilm formation and the production of hydrolytic enzymes such as proteases, phospholipases, and hemolysins, which contribute to the adherence, cellular damage, and tissue invasion in the host [65,143,146,147,148]. In addition, they can form a germ tube and they possess phenotypic and genotypic variability [142]. Such virulence factors grant them the ability to evade the host’s immune response and generate antifungal resistance [147,149].

Among the virulence factors reviewed in this work are those involved in the adherence and biofilm formation mechanisms (Figure 2).

### Adherence Mechanisms

The adherence mechanism is an important virulence factor that is regulated by diverse genes and marks the beginning of infection by *Candida* spp. [150]. Adhesion to host cells is essential for any commensal pathogen, as it allows a firm adherence and avoiding being dragged for elimination [151]. Therefore, it is relevant for the establishment and persistence of the disease [146]. *C. albicans* and *C. glabrata* have independently developed specific adhesins, requiring specific or non-specific receptors, as well as various host signals. When the "host" environment is detected, an expression of adhesins occurs that allows binding to the cell receptor [151].

It has been observed that various factors of the environment in which it is located favor adherence of the *C. glabrata* complex to host cells. Among them is increased acidity in the environment and extracellular polymer production by bacteria located in the oral mucosa of patients using dental prostheses. In other cases, elevated levels of estrogen and glycogen in vaginal secretions during pregnancy provide a carbon-rich nutritional source for these yeasts. Likewise, other studies suggest that the presence of *C. albicans* seems to improve *C. glabrata* complex strains’ adherence by injuring mucous membranes and moderately rough and hydrophobic surfaces that create a favorable niche for the microorganism colonization [150,151,152,153].

The hydrophobic cell wall of *C. glabrata* is the place where the physicochemical interactions between the yeast and the colonized region occur. The wall has a surface layer composed of glycoproteins that contributes to host cell recognition and contains various chitin and glucan chains extended along the entire cell wall structure. They present a β-(1,3)-glucan, β-(1,6)-glucan branched and bound to chitin by a β-(1,4)-glucan bond, as well as various adhesin-type cell wall proteins (CWP) [150,154,155,156,157,158,159].

Epithelial adhesins, or *Epa* proteins, are the main cell surface proteins involved in *C. glabrata* virulence [65,154,160]. *Epa* proteins are analogous to *Als* proteins in *C. albicans* and thus are important for the adherence mechanism. *Epa* genes are responsible for encoding these proteins [39,146,156,161]. Gabaldon et al. [162] identified an expansion of the *Epa* gene family that only appears in the three pathogenic species of the *Nakaseomyces* clade: *C. glabrata*, *C. nivariensis,* and *C. bracarensis*.

There is great variability in the number of *Epa* genes among *C. glabrata* complex isolates. It is considered that this gene family is composed of 17 to 23 members, depending on the isolate. Sequencing of the CBS138 *C. glabrata* strain revealed the presence of 18 genes, while the BG2 strain only has 23 *Epa* genes. *C. bracarensis* shows 12 genes and *C. nivariensis* only 9; while the non-pathogenic strain *Nakaseomyces delphensis* only has one *Epa* adhesin. Therefore, *Epa* genes are vital virulence factors in the *C. glabrata* complex [39,50,85,152,158], and their presence could explain the highly pathogenic potential observed in *C. nivariensis* and *C. bracarensis* [78,162].

Epithelial adhesin 1 (*Epa1*) has been identified as the only one needed for *C. glabrata* to bind to the epithelial surface of the host in vitro as it regulates the interaction between yeast and the host’s epithelial cells [158]. In addition, it is involved in biofilm formation [163] and participates in the adherence mechanism in immunological, endothelial, and epithelial cells. It is activated by proteolysis through *C. glabrata* yapsins (*CgYps*), a family of glycosylphosphatidylinositol aspartyl proteases involved in protein maturation and cell wall remodeling, maintenance, and preservation. This type of *yapsins* (*YPS*) is encoded by the twelve *CgYPS* genes [158,164].

In vitro studies have shown that *Epa1* gene deletion reduces adhesion to host epithelial cells, suggesting it has a crucial role in fungal adhesion to abiotic substrates [78]. It has also been observed to be regulated by the *Pdr1* transcription factor [65,118,142,158,165].

The *EPA6* and *EPA7* genes are functional adhesins present in upper and lower urinary tract infections. Although the *EPA6* gene has not been expressed in in vitro studies, it has been observed during urinary tract infections [48,63,65,164]. This contributes to the hypothesis that *C. glabrata* can adapt to adverse environmental conditions improving its adherence due to these genes [48,63,65,164].

*Epa* gene expression is complex and is regulated by subtelomeric silencing based on chromatin and transcriptional factors. Subtelomeric silencing is composed of the *SIR* complex (*Sir2, Sir3,* and *Sir4*), which requires NAD+ as a cofactor, *Rap1* that recruits the *SIR* complex, as well as *Rif1*, *yKu70,* and *yKu80,* which may respond to various environmental factors [85,154,156,158,166].

Moreover, certain proteins have been identified in *C. glabrata,* such as *Pwp7p* and *Aed1p*, that are relevant for adherence in endothelial cells and are not observed in *S. cerevisiae* nor *C. albicans*. Such proteins are anchored to glycosylphosphatidylinositol (GPI), and it has been observed that the genes encoding their synthesis regulate the ability of *C. glabrata* to adhere to the epithelial tissue after facilitating binding to carbohydrates present in the host cell. Therefore, they are relevant for yeast virulence [150,154,156]. One study showed that the adherence was reduced by 66% in isolates with a mutation in the *Pwp7p* protein, and those with an *Aed1p* mutation had a 50% reduction in their adherence capacity [167].

The adherence capacity of *C. glabrata* complex yeasts is relevant for the development and persistence of infections in humans as it leads to biofilm formation on biotic surfaces and, especially, on abiotic ones [67,163]. Figueiredo-Carvalho et al. [168] noted that *C. nivariensis* adheres to inert surfaces, showing a predilection for polystyrene. Other studies have shown that *C. glabrata* has a greater tendency to adhere to acrylic surfaces in dentures, predisposing patients who need to wear them to suffer from oral candidiasis [169]. Furthermore, as already mentioned, this yeast has a greater adherence capacity to urinary epithelium cells compared to other *Candida* non-*albicans* species [170]. Nonetheless, Vieira de Melo et al. [171] reported seven *C. glabrata* isolates with low adherence capacity to oral epithelium cells compared to other *Candida* spp. [171].

## 6. Enzymatic Activity

Hydrolytic enzymes facilitate *Candida* spp. Adherence, facilitating the yeast’s penetration and invasion into the host tissues causing infection [9].

In *C. glabrata*, the production of hydrolytic enzymes (proteases, phospholipases, and lipases), and their release to the local environment, causes destruction of the host tissues, including skin, vaginal, and oral mucosal membranes [146].

Riceto et al. [142] observed that some *C. glabrata* isolates lack virulence factors, such as phospholipase, proteinase, and DNAse activity. This coincides with the observations of Mutlu et al. [172], who reported the absence of biofilm formation, phospholipase, proteinase, and esterase activity in two *C. glabrata* isolates collected in an Intensive Care Unit (ICU) in Turkey.

In *C. bracarensis*, the presence of aspartyl protease, phospholipase, hemolysin, and catalase activity has been observed, with no DNAse and coagulase activity [63]. Moreira et al. [173] also reported proteinase and hemolysin activity in this yeast.

### 6.1. Proteases/Proteinases/Aspartyl Proteases

Among the best-known *C. glabrata* virulence factors are the aspartyl proteases or yapsins that belong to the *YPS* family and are encoded by 12 genes [139]. These proteins significantly increase the microorganism’s ability to survive within human macrophages and play an important role in cell wall remodeling by removing and releasing proteins anchored with glycosylphosphatidylinositol [174,175].

Proteinases allow the colonization and invasion of host tissue through ruptures in mucous membranes, and act by degrading immunological and structural defense proteins [65,175] such as heavy-chain IgG, alpha2-macroglobulin, protein C3, beta-lactoglobulin, lactoperoxidase, collagen, and fibronectin [65]. Likewise, there are studies that demonstrate the proteinases’ ability to degrade a protein substrate, thus suggesting their pathogenic role in infections caused by *Candida* spp. [176].

Swoboda-Kopeć et al. confirmed that the *YPS2*, *YPS4,* and *YPS6* genes prevailed in most strains of *C. glabrata* isolated from clinical samples, and that the prevalence of the same genes in *C. nivariensis* was low [56].

Kaur et al. in their study observed that mutant strains of *C. glabrata* who had *YPS* gene nominations did not show a virulent phenotype [138].

*C. albicans* has been shown to be the species with the highest production of proteinases in various studies carried out [145,148,176]. Some studies show that *C. glabrata* display activity of an important proteinase, which positions it among third [176] and fourth [145] in activity after species like *C. albicans*, *C. tropicalis,* and *C. krusei* [145,176]. Atalay et al. also reported the presence of activity of proteinases in strains of *C. glabrata*, observing it in 28% [148], while Subramanya et al. reported strong proteinase activity in one of nine isolates of *C. glabrata* [95].

Furthermore, there are also studies such as that of Hacioglu et al. (nine isolates) and that of Rossoni et al. (four isolates) that do not report the activity of proteinases in their isolates of *C. glabrata* [177,178]. A more recent study by Sriphannam et al. coincides in which the six strains of *C. glabrata* analyzed, did not show a proteinase activity [179]. The study by Barbosa et al. also showed an absence of proteinase activity in 3 of 4 isolates of *C. glabrata*, where the remaining isolate showed moderate activity [180].

The study carried out by Pereira et al. in a group of 50 healthy patients and another 50 patients with stomatitis and different types of lesions showed that the isolates of *C. glabrata* from healthy patients (12 isolates) had moderate proteinase activity in 17%, and the remaining 83% showed no such activity; those from patients with stomatitis (24 isolates) and type I lesions showed moderate activity in only 20%, and 80% were without activity; patients with type II lesions showed moderate activity in 7% and activity was not observed in 93%; finally, patients with type III lesions showed moderate activity in 50%, while the remaining 50% did not show proteinase activity, and this indicates that in most isolates of type III lesions, the production of virulence factors was higher than for the rest of the groups [181].

With respect to the other two species of the *C. glabrata* complex, there are also few studies that evaluate the presence of enzyme activity due to its poor molecular identification. Treviño-Rangel et al. studied the enzymatic activity of *C. bracarensis*, which presented very strong activity of aspartyl proteinase [63]. Moreira et al. also studied the enzymatic activity of three strains of *C. bracarensis* and confirmed that the strain isolated from a central venous catheter in a hospital in the United Kingdom presented proteolytic activity, that is, released proteases (aspartyl proteases) to the culture medium [173]. In the case of *C. nivariensis*, Tay et al. confirmed the absence of proteinase activity in two strains [182]; however, Figueiredo-Carvalho et al. analyzed one strain of *C. nivariensis* from a runny nose and observed elevated protease activity [168]. Finally, the most recent study conducted by Hernando-Ortiz et al. with strains of *C. glabrata*, *C. nivariensis,* and *C. bracarensis* (two strains for each species) showed that none of the strains had proteinase activity [62]. The above observations exposed by the researchers and their working groups indicate that not all strains of this complex present enzymatic activity, with some strains more virulent than others.

### 6.2. Phospholipase

Phospholipases are hydrolytic enzymes with direct action against phospholipids, therefore generating damage to the cell membrane, in addition to potentiating the invasion of the mucosal epithelium in the host [9,180,183]. After disruption of the epithelial cell membranes, the tips of the hyphae penetrate the cytoplasm resulting in cell lysis and tissue damage [9,180,183].

Several studies have reported phospholipase activity in species such as *C. albicans*; however, in *C. glabrata,* this activity is not frequent [179]. Of the studies carried out to evaluate the activity of phospholipases in the *C. glabrata* complex, the vast majority conclude the absence of such enzymatic activity [62,168,176,177,182,184]. The study conducted by Kumari et al. reported strong enzymatic activity phospholipases in three of its strains corresponding to *C. glabrata*, which corresponded to 18.75% of a total of 16 strains from women with vulvovaginal candidiasis in India [185].

Likewise, the study of Pereira et al. evaluated patients with prosthetic stomatitis and varying degrees of injury in healthy individuals. Twelve healthy patients showed a positive culture for *C. glabrata*, presenting moderate activity of phospholipase in 25% and strong activity of the same in 75%; the group of patients with stomatitis and type I lesions showed moderate phospholipase activity in 40% and strong activity in 60%; patients with type II lesions showed moderate and strong activity in 20% and 80%, respectively, and 100% of patients with type III lesions showed strong phospholipase activity [181]. Furthermore, another study showed the production of phospholipase in different isolates of *Candida* spp., of which six of seven belonged to *C. glabrata* [171].

Kalaiarasan et al. observed that 15/51 (29.41%) of their *Candida* spp. isolates showed phospholipase activity; however, only one isolate (4.3%) corresponded to *C. glabrata* [145].

A study conducted in Nepal with 71 isolates of Candida spp. showed that only nine of them were *C. glabrata* (12.67%), and only three isolates presented phospholipase activity. In Turkey, from 50 isolates, 14 belonged to C. glabrata (28%), and only 5 strains showed phospholipase activity. Similarly, a study in Turkey with 100 isolates of *Candida* spp. showed only 9 *C. glabrata* isolates (9%) and 3 strains with phospholipase activity [95,148,178].

With respect to *C. nivariensis* and *C. bracarensis*, the studies conducted have reported an absence of phospholipase activity in both species [62,173]. However, Treviño-Rangel et al. observed very strong activity of phospholipase in *C. bracarensis* in their study [63].

### 6.3. Esterase

Some studies suggest that the virulence of *Candida* spp. species is due to the toxic effects caused by both lipases and esterases on host tissues [186]. These proteins act by degrading the ester bonds of cell membranes by increasing cell invasion [138]. The activity of esterases has been observed in very few isolates of the *C. glabrata* complex, and in some studies, it has not been reported [63,95,145,148,168,178,179].

Kalaiarasan et al. observed that 30.4% of *C. glabrata* strains presented esterase activity, compared to *C. tropicalis* with 66.7% and *C. albicans* with 33.3% [145].

Atalay et al. reported the production of esterase in one of fourteen strains of *C. glabrata*, with this being the species with the lowest production in that study [148]. Sriphannam et al. showed that two of their six strains tested for *C. glabrata* had medium-level esterase activity [179]. In the study conducted by Hacioglu et al., they were also only able to identify esterase activity in a strain of *C. glabrata* [178]. Likewise, the study of Subramanya et al. observed strong activity in three of nine strains of *C. glabrata* [95]. Studies have reported the absence of esterase activity in both *C. nivariensis* as in *C. bracarensis* [63,168].

### 6.4. Hemolysin

Hemolysins present phospholipases type C activity and affect the membrane of erythrocytes and lymphocytes, generating the rupture of erythrocytes and cellular damage in lymphocytes, facilitating the recovery or extraction of elemental iron present in host cells, allowing the survival and persistence of the microorganism [65,187]. Hemolytic activity tends to be different in different types of *Candida* spp. [188].

In most of the studies carried out, it has been observed that the species of the *C. glabrata* complex that usually produce hemolysins in large quantities are *C. nivariensis* and *C. bracarensis*, since those made in *C. glabrata* report the absence of such activity. Luo et al. observed that all *C. glabrata* strains (33 isolates) were able to produce hemolysins in vitro (100%), and that the gene of the protein similar to hemolysin (HLP) is related to hemolytic activity in this yeast [189].

There are studies that show that *C. glabrata* has poor hemolytic activity as well as an inability to import heme efficiently compared to *Candida albicans*, so it is suggested that *C. glabrata* depends mainly on the circulating sources of Fe in the host to be able to meet its needs for this micronutrient [190].

Similarly, Kalaiarasan et al. observed the production of hemolysins in 87% (20/23 strains) of the *C. glabrata* strains analyzed [145]. Vieira de Melo et al. also reported the production of hemolysins in all strains (7/70 strains) analyzed in their study and observed that the *C. glabrata* strains had higher production than the *C. albicans* isolates and other species [171].

The study of Riceto et al. observed that strains of *C. glabrata* had moderate hemolytic activity [142]; the above matches our most recent study where moderate and strong hemolysin activity was reported in both healthy patients and patients with stomatitis and different types of lesions [181].

A 2016 study showed the absence of hemolysin in a strain of *C. nivariensis* [168]. However, a more recent study showed that strains of *C. nivariensis* and *C. bracarensis* produced alpha or partial hemolysis, contrary to *C. glabrata,* which showed gamma hemolysis; that is, there was no hemolysis [62].

In the case of *C. bracarensis*, a study showed that three strains tested had total hemolytic activity when found in a medium of agar sheep’s blood enriched with glucose [173]; in another study with a strain of *C. bracarensis* in Mexico, very strong hemolysin activity was also reported [63].

## 7. Biofilm Formation

Species of the genus *Candida* usually adapt to different environments, forming microbial communities that irreversibly adhere to surfaces (inert material or living tissue) called biofilms [65]. The formation of biofilms in species of *Candida* is a recent clinical problem and is associated with a higher mortality rate in patients with infections caused by these pathogens [191,192]. It is considered to be the most prevalent form of growth in microorganisms [192].

Therefore, the formation of biofilms is one of the most important virulence factors for pathogenicity in species of *Candida* [21,147,186] and is associated with recurrent infections and treatment failures [163,193]. The ability of the isolates of *Candida* spp. to form biofilms varies according to the species studied [21,163,194,195,196].

The adhesion mechanism will result in the development of biofilms, which will provide a favorable and protective environment for the growth of the members of the *C. glabrata* complex [157].

Its formation comprises, in the first place, the adhesion and colonization of an abiotic and/or biotic surface, followed by cell division or proliferation forming a basal layer of anchoring microcolonies, culminating in the maturation of the biofilm that involves the formation of filaments, hyphae, and/or pseudohyphae as well as extracellular matrix production [191]. The extracellular matrix is rich in carbohydrates and proteins (especially in *C. glabrata*) [173]; it has the function of protecting cells and acting as a barrier against drugs and other substances toxic to the microorganism, keeps nutrients inside to reach biofilm cells [191], and contributes to intrinsic resistance to the host’s immune system and other environmental alterations [72]. When the biofilm is mature, it has the ability to detach and disperse, [191] and subsequently, it can colonize new sites, thus completing its life cycle [83].

Several studies agree that the formation of biofilms occurs more frequently in species of *Candida* non-albicans [72,145,171,178,179,185,186,195,197]; however, the production of biofilms by the *C. glabrata* complex seems to be absent [148,176], but if it does occur, it is usually mild to moderate production [145,171,197] and some studies report that its production increases when it is co-cultivated with species such as *C. albicans* [163].

It has been observed that the biofilms of *C. glabrata* consist of a compact monolayer or multilayer that does not form filaments [65,163,196]. Kraneveld et al. identified seven *Awp* adhesins (*Awp 1**–7*) [198], a family of adhesins previously identified by Groot et al. (*Awp 1**–4*) and is suggested to be involved in the first stage of the development of biofilms in *C. glabrata* [199].

The formation of biofilms has been closely linked to the development of antifungal resistance in the *C. glabrata* complex [29,139,192] and involves the participation of various genes and the control of their expression by complexes such as *Sir* (*Sir2-Sir4*) and *Swi/Snf,* which in *C. glabrata* seem to be the basis of regulation for the formation of biofilms [192] since they are involved in adhesion of the pathogen [167,200].

Santos et al. reported in their study that multi-drug resistance transporters (*MFS*) *CgTPO1_1* and *CgTPO1_2* are important for virulence of *C. glabrata* since its deletion was associated with an increase in survival of the *Galleria mellonella* infection in vivo [201]. The study also found that *CgTPO1_2* is positively regulated during biofilm formation [201]. Again, Santos et al. managed to identify that *CgTec1* (an ortholog of *CaTec1* in *C. albicans* and the main regulator of its biofilm formation) was necessary for the activation of the transcription of four *MFS*, *CgTPO1_2*, *CgQDR2*, *CgTPO4,* and *CgDTR1,* in the early stages of biofilm formation. They concluded that then, the four *MFS* act to the benefit of the microorganism since their deletion caused a significant decrease in the formation of biofilms in *C. glabrata*; likewise, it was observed that the delegation of *CgDTR1*, *CgTPO4,* and *CgQDR2* increases the potential of the plasma membrane, leading to a decrease in the expression of genes encoding adhesins such as *CgALS1* and *CgEPA1* during the formation of biofilms, significantly decreasing them [202].

One study showed that the *CgFab1*, *CgVac7,* and *CgVac14* signaling components for phosphatidylinositol 3,5-bisphosphate (*PI (3,5) P2*) genes are important for biofilm formation, cell survival, and virulence of *C. glabrata*, since strains with mutations in these genes tend to have a defective cell wall and because the impaired vacuolar functions and biofilm formation observed in vitro was diminished [203].

The study of Kalaiarasan et al. reported that *C. glabrata* was ranked third in terms of biofilm production with seven strains (30.4%); two of the strains had moderate production, and the remaining five, mild production of biofilms [145]. Although other studies such as Pongrácz et al. also observed increased production of biofilms in *Candida* non-albicans, only two strains of *C. glabrata* showed biofilm formation [197]. The data also match what was observed by Sriphannam et al. where *C. glabrata* corresponded to 33% biofilm production, of which two had a high production, two were moderate, one was low, and the rest showed no production [179].

The strains analyzed by Vieira de melo et al. showed low production of biofilms corresponding to *C. glabrata* [171]. In the study of Hacioglu et al., all species of *Candida* non-*albicans* formed biofilms; nine of the strains corresponded to *C. glabrata* [178]. Likewise, Subramanya et al. reported the production of biofilms in six of nine *C. glabrata* strains analyzed [95].

Gonçalves et al. compared the formation of biofilms of *C. albicans* and *C. glabrata* in an acidic environment (pH of 4) similar to the vaginal environment, as well as a neutral environment. They noted that *C. glabrata* formed thicker biofilms under an environment with acidic conditions than in a neutral environment (contrary to what was observed in *C. albicans*), which suggests that the microorganism presents an adaptability to the acidic environment of the vagina as another virulence factor, explaining its presence in women with recurrent vulvo-vaginal candidiasis (CVV) [204]. Likewise, they observed a greater number of components of the biofilm matrix (proteins and carbohydrates) under neutral conditions than under acidic conditions, except for a strain of *C. glabrata* where the opposite was observed [204]. The findings coincide with a study conducted by Gonçalves et al., where they analyzed the effect of progesterone and β-estradiol on the production of biofilms in *C. glabrata* and *C. albicans*. They noted that hormones do not influence the generation of biofilms in C. *glabrata*, but they reduce biofilm formation in *C. albicans* by more than 65% [205].

Finally, they suggest that hormones act as environmental signals that promote the protection of *Candida* spp. [205]. This is related to what Beyer et al. described when they observed that CgHog1, a high-osmolarity glycerol response MAP kinase that contributes to *C. glabrata* persistence inside mice macrophages, may be the main quantitative determinant of lactic acid stress resistance. Thus, they suggest that CgHog1 is important for *C. glabrata* survival in the common vaginal microbiota as it allows it to tolerate different *Lactobacillus* species [84].

On the other hand, studies such as those from Atalay et al. (9/50 strains) and Yagmur et al. (12/99 strains) do not report biofilm formation in any of the *C. glabrata* isolates analyzed [148,176].

One study showed that *C. glabrata* can produce a higher biomass of biofilm on silicone surfaces and in the presence of urine, unlike other species [206]. One study found that the biofilm biomass produced by *C. glabrata* is greater than in *C. nivariensis* [64]. Moreira et al. demonstrated biofilm production in the three studied *C. bracarensis* strains, and they all showed differences in their extents [173].

## 8. Conclusions

The various virulence factors and resistance mechanisms to antifungals presented by species of the *C. glabrata* complex contribute to the perfect pathogenic combination that has allowed this yeast to become one of the most frequent agents of candidiasis.

## Figures and Tables

**Figure 1 pharmaceutics-13-01529-f001:**
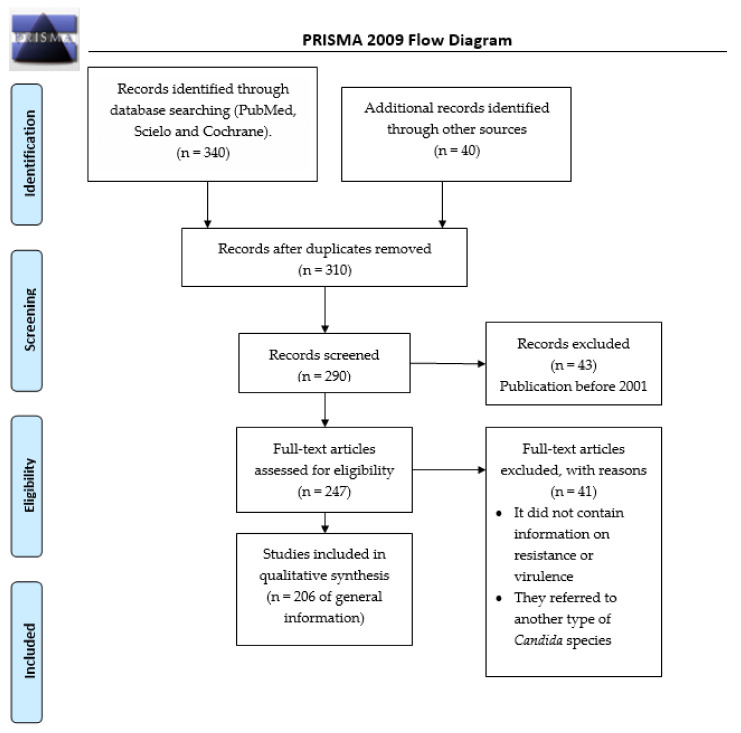
Flowchart of the different phases of the systematic review.

**Figure 2 pharmaceutics-13-01529-f002:**
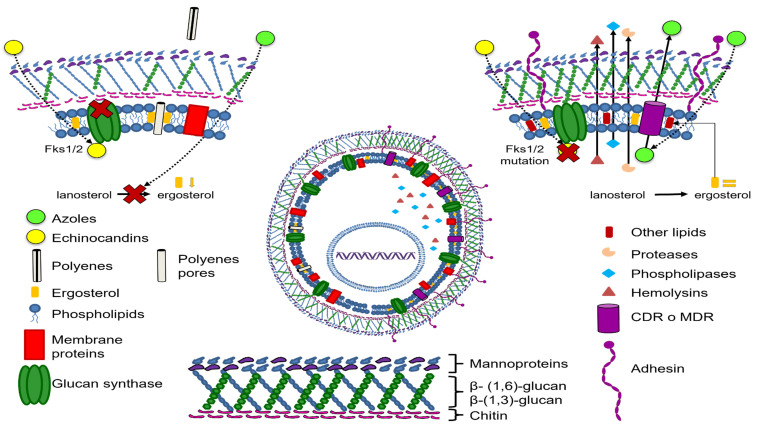
*Candida glabrata* drug resistance and virulence mechanisms.

**Table 1 pharmaceutics-13-01529-t001:** Susceptibility variations of *Candida glabrata* complex per continent.

Continent	Country	Number of Strains	Susceptibility and Resistance				References
			Azoles	Echinocandins	Polyenes	Other	
Africa	Ethiopia	17	Clotrimazole R: 11.76% Itraconazole: R: 76.47% Ketoconazole: R: 41.18%				[14]
	Cameroon	33	Fluconazole: R: 3.03%				[104]
	Ghana	31	Fluconazole: S: 19.4% ISDD: 41.9% R: 38.7% Voriconazole: S: 54.8% ISDD: 9.7% R: 35.5%		Nystatin ISDD: 12.9% R: 87.1%		[105]
America	Canada	392	Fluconazole: ISDD: 87.8%				[37]
	Chile	37	Fluconazole: R: 6.6% Itraconazole R: 20%	Micafungin R: 10%	Amphotericin B R: 2.7%		[16]
	Brazil	38 Susceptibility tests were performed in 8	Fluconazole R: 50% Miconazole R: 12.5%		Amphotericin B S: 100% Nystatin S: 100%		[51]
Asia	China	73	Fluconazole R: 6.8% Voriconazole R: 6.8%		Amphotericin B R: 100% 5-fluorocytosisin: R: 11%		[20]
	India	21	Fluconazole S: 100% Itraconazole S: 100% Posaconazole S: 100% Ketoconazole S: 100% Voriconazole S: 100%	Caspofungin S: 100%	Amphotericin B S: 100% 5-fluorocytosisin S: 100%		[21]
	India	22	High Resistance to clotrimazole, fluconazole, itraconazole, ketoconazole: 100%		High Resistance to nystatin: 100%		[70]
	Iran	4 (*C. nivariensis*)	Fluconazole S: 100% Itraconazole S: 100% Voriconazole S: 100% Posaconazole S: 100%	Micafungin S: 100%	Amphotericin B S: 100% 5-fluorocytosine S: 100%		[106]
	Nepal	9	Fluconazole S: 66.6% ISDD: 11.1% R: 22.2% Voriconazole: ISDD: 88.8% R: 11.1%	Caspofungin S: 100%	Amphotericin B ISDD: 100%		[95]
	Kuwait	11	Fluconazole: R: 64% ISDD: 36%				[107]
	Kuwait	75	Fluconazole: R: 48% ISDD: 52%	Micafungin: R: 4% ISDD: 2.67% Caspofungin: R: 5.33%	Amphotericin B R: 6.67%		[108]
	Turkey	12	Fluconazole R: 100%				[109]
	Turkey	83	Fluconazole: R: 9.2% Itraconazole: R: 45.8% Voriconazole R: 43.4%				[71]
Europe	Spain	14	Fluconazole: ISDD: 100% Itraconazole: R: 50% Voriconazole: S: 100%	Caspofungin: S: 92.9% ISDD: 7.1% Micafungin: S: 100% Anidulafungin: S: 85.7% ISDD: 14.3%	Amphotericin B S: 100% 5-fluorocytosisin: S: 100%		[101]
	Ireland	21	Fluconazole: R: 37%		Amphotericin B: R: 14%		[5]
	Poland	445	Fluconazole: ISDD: 100% Itraconazole: R: 41% Posaconazole: R: 50% Voriconazole: S: 83%	Caspofungin, anidulafungin and micafungin: S: 100%	Amphotericin B: S: 100%		[57]
	Greece	34	Fluconazole: ISDD: 94% R: 6%	Anidulafungin: S: 97% R: 3% Caspofungin S: 88% ISDD: 3% R: 9% Micafungin S: 97% R: 3%			[24]
	Germany	176	Fluconazole: R: 38%	Anidulafungin: S: 52% R: 48%		Combined resistance to fluconazole and echinocandins: 14%	[110]
	Switzerland	5	Fluconazole: ISDD: 100%	Caspofungin, anidulafungin and micafungin R: 100%			[111]
	United Kingdom	7225		Caspofungin, anidulafungin and micafungin: R: 0.55%			[112]
	Poland	81	Fluconazole: R: 22.2% ISDD: 77.7% Fluconazole and voriconazole: R: 1.2% Voriconazole R: 7.4% Cross-Resistance to other azoles: R: 18.5%	Caspofungin, anidulafungin and micafungin: S: 100%	Amphotericin B: S: 100% 5-fluorocytosine: S: 93.8% ISDD: 3.7% R: 2.5%		[113]
	Poland, France, Greece, Germany, Italy, Czech Republic, Spain, Austria, Serbia, Iran, India, Thailand, United States	64	Fluconazole: R: 1.6% Itraconazole: R: 1.6% Isavuconazole S: 100% Posaconazole: R: 3.1% Voriconazole R: 3.1%	Caspofungin, anidulafungin and micafungin S: 100%	Amphotericin B: S: 100% 5-fluorocytosine: S: 100%		[114]
	Germany Spain	4 1	Fluconazole: R: 100% Voriconazole: R: 100%		Amphotericin B: R: 100%		[115]
	Jerusalem	176	Fluconazole: ISDD: 81.25% R: 4% Voriconazole: R: 4.7%	Caspofungin: R: 33.6%	Amphotericin B S: 100%		[11]
Oceania	Australia	35	Fluconazole: R: 22.8%	Caspofungin: R: 17.1%			[3]

S: Susceptibility, ISDD: Intermediate Susceptibility Dose-Dependent, R: Resistance.

**Table 2 pharmaceutics-13-01529-t002:** Drug resistance fluctuations caused by the type of Candida and genetic variations.

Yeast	Drug on Which Resistance Is Generated		Antifungal Resistance			References
			Genes and Proteins Involved	Mechanisms Involved	Result	
Complex *C. glabrata*	Azoles	Multiple drugs	Mutations (polymorphisms, deletions, etc.) in the *Pdr1* gene of the PDR1 transcription factor and TAC1 transcription factor	Overexpression and activation of ATP-binding cassettes [CDR1, CDR2 (also designated PDH1, SNQ2, *FAA1*)]	Drugs transported to the outside of the cell	[8,113,117,118,119,120,121,122,123,124,126]
					Decreased cell surface hydrophobicity during biofilm formation	[125]
					Modification of biological transport pathways of hydrophobic compounds and lipid metabolism	[126]
	Azoles		Mitochondrial dysfunction associated with the development of mitochondrial DNA-deficient “small mutants.”		Drugs transported to the outside of the cell	[64,120,124]
	Azoles		Deletion of *CgADA2*	Positive regulation of adherence factors	Thermotolerance and hypervirulence	[127]
	Echinocandins		Mutations in *FKS1* and *FKS2* in the hot spots *HS1* and *HS2* (Examples: *FKS1 S629P, FKS2 F659 FKS2 S663P* y)	Altered conformation of the 1,3-β-glucan-synthase subunits Fks1p and Fks2p	Reduced affinity of echinocandins for β-1,3 glucan.	[108,110,111,129,130,131,132,133]
	Azoles/polyenes		Mutations in the ERG6 and *ERG11* proteins	Alteration of sterol 14-α-demethylase	Less ergosterol content with cell membrane modification.	[134]
					Ergosterol exchanged for generated exogenous sterols	[115]
*C. nivariensis*	Azoles		Increased expression of YPS1, AWP3, EPA1, ERG11, CDR1, and CDR2 genes		Antifungal resistance and increased virulence	[86]
			Increased mRNA expression of ERG11, CDR1, and CDR2	Overexpression and activation of CDRs	Increased antifungal resistance	[102]

## Data Availability

Not applicable.

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
