# Peer review of "Candida glabrata Antifungal Resistance and Virulence Factors, a Perfect Pathogenic Combination"

_pharmaceutics, 2021, doi:10.3390/pharmaceutics13101529_

Round 1
Reviewer 1 Report
Review of the article: “Candida glabrata antifungal resistance and virulence factors, a perfect pathogenic”
Submission ID - pharmaceutics-1339297
In my opinion this manuscript is a good starting point for preparing a really interesting review article. However, the current version cannot be accepted. First of all the authors should ask a native speaker for revision of this text – I am not a specialist in English grammar, but in some places I had some problems with understanding the idea of the authors. Moreover, the authors should eliminate all typing errors from the text (e.g. some capital letter). My final decision is major revision. Below I have presented some detailed comments – most of them are of minor importance.
Abstract – The abstract is interesting and well prepared – no important critical remarks. However, the authors should use full names of microorganisms when they are first time presented in the text of publication (the same comment for introduction).
Introduction – Azoles are currently the “most important” and most commonly used antifungal agents. In line 87 the authors wrote: “ … as they exhibit greater resistance to azoles”; in my opinion it should be emphasized that this greater resistance to azoles and overuse of these substances are the most important reasons of growing frequency of infections caused by C. glabrata and other non-albicans candida species.
Materials and Methods
I am confused about the period of time that was considered. In line 106 (abstract) it is between 2010 and 2021 and in the section MM the authors mention completely another period of time (from January 2020 to February 2021). Moreover in the graph the authors mentioned that they excluded paper older than 20 years of publication. Another information presented in Fig. 1 that is not clear for me is the reason of excluding of 43 publications – why were these publications excluded?
The legend to Fig. 1 should contain more information.
Section 3
Line 123 – beta-glucan and chitin are components of the cell wall not cell membrane – the same comment for lines 134
Lines 140 and 141 – TNF is not produced in C. glabrata
Lines 151-152 – the authors have presented a set of genes and several proteins encoded by these genes but it should be written which gene is responsible for production of particular protein.
Lines 178-179 – this sentence (can obtain free iron or iron from proteins and promote resistance 178 to the death of macrophages using the siderophore-iron transporter Sit1) is not clear for me.
Lines 192 – 201 – this part is not clear for me (e.g. dependence between IL-1β production and CgYapsins should be explained).
Section 4
Line 204 – could the authors give some examples of isolation of C. glarbarat resistant to polyenes? Because of the mode of action resistance to this class of antibiotics is rarely observed.
Line 216 “due to their low cost” – in my opinion there are many other important advantages (more important than low cost) of azoles, including low toxicity and good water solubility (fluconazole) – it must be mentioned in scientific paper.
If authors are cited (e.g. lines 222 and 225) the year of publication also should be provided
Lines 227 – 232 – but how many strains from Ghana were investigated in this study (strains were isolated from 176 pregnant patients – does it mean that there were 176 isolates?). I have some doubts about results concerning resistance to nystatin (but of course it is not a critical remark to the authors of this publication).
Line 312 – why the authors have presented this form of citations ((Vermitsky and Edlind, 2004).)
The authors performed detailed analysis of publications concerning epidemiology, resistance and virulence of C. glabrata – from my study of literature I know that there are many other publications concerning investigation of C. glabrata isolated in Europe (I am mostly interested in this region) some of them should be cited. In some of these publications authors also investigated mechanisms responsible for resistance.
Short comment to information presented between lines 304-327 (overexpression of drug transporters) – interesting results in this area were presented by Gucwa and co-workers (2015 J Med Microbiol) who observed up regulation of CDR1 gene as a reason of azole resistance among C. glabrata strains isolated in Poland. Up regulation of CDR2 and ERG11 was not observed . Moreover the authors did not observe any mutations within EGR11 gene – these mutations are identified as important mechanism of azole resistance among C. albicans.
Line 352 - this sentence is not clear for me - “Pan drug resistance to these antifungals”
Lines 359 – 363 and 365 – 371. I would be grateful for a short comment. In both cases deletion of target genes resulted in increase of susceptibility. But is this the proof that proteins encoded by these genes are crucial for resistance?
Line 393 – gene’s delegation? (not clear), beside of that which gene?
Lines 397-400 – how many strains were investigated?
Section 5
Line 427 – the title of Figure 2 should be translated (it is not in English)
Line 432 – it is relevant
Line 447 – why the cell wall is hydrophobic (it is composed of “carbohydrates”)
Section 6
Line 513 – do the authors mean any particular tissue? I would propose using plural in this place.
Line 547 – 548 – how many strains were investigated in these studies
Line 549 – why capital letter
Lines 551-560 – the percentage is not important if we do not know the number of isolates.
Line 563 why capital letter (They??) – the same problem in line 564
Line 570-572 – again, it is important to know the number of strains tested
Line 599 – it is too general information
Lines 634-639 – the authors must give the number of strains tested (percentage is useless if we do not know the number of isolates that were considered in these studies)
Section 7
Lines 742-743 – how many strains were investigated
Final decision – major revision
Author Response
Answers to Reviewer 1 concerns:
In my opinion this manuscript is a good starting point for preparing a really interesting review article. However, the current version cannot be accepted. First of all, the authors should ask a native speaker for revision of this text – I am not a specialist in English grammar, but in some places, I had some problems with understanding the idea of the authors. Moreover, the authors should eliminate all typing errors from the text (e.g., some capital letter). My final decision is major revision. Below I have presented some detailed comments – most of them are of minor importance.
Answer: We are thankful for the time and effort you have invested in the revision of our manuscript. All your suggestions have enriched our work. In the manuscript, the additions are highlighted in yellow. Please find our answers to your valuable recommendations; we hope that we have addressed all your concerns. The manuscript has been reviewed by an expert in the English language.
Abstract – The abstract is interesting and well prepared – no important critical remarks. However, the authors should use full names of microorganisms when they are first time presented in the text of publication (the same comment for introduction).
Answer: Thanks for the observation made, we corrected using the full names of the microorganisms for the first time within the abstract and the introduction
Introduction – Azoles are currently the “most important” and most commonly used antifungal agents. In line 87 the authors wrote: “as they exhibit greater resistance to azoles”; in my opinion it should be emphasized that this greater resistance to azoles and overuse of these substances are the most important reasons of growing frequency of infections caused by C. glabrata and other non-albicans candida species.
Answer: Considering your opinion, the paragraph was modified in the introduction, lines 88-91 and was as follows:
“It should be noted that the increasing use of azole antifungals for the treatment of superficial and systemic infections by Candida glabrata has led to the selection and emergence of resistant isolates, as well as increased infections by other non-albicans species”
Materials and Methods
I am confused about the period of time that was considered. In line 106 (abstract) it is between 2010 and 2021 and in the section MM the authors mention completely another period of time (from January 2020 to February 2021). Moreover, in the graph the authors mentioned that they excluded paper older than 20 years of publication. Another information presented in Fig. 1 that is not clear for me is the reason of excluding of 43 publications – why were these publications excluded?
Answer: We appreciate the comment and both in the abstract and in materials and methods it was corrected the interval of years from 2001 to 2021.
Line 38 in the abstract
Line 108 in materials and methods
In PRISMA or figure 1, 43 publications were excluded because they were published before 2001.
The legend to Fig. 1 should contain more information.
Answer: According to your comments, in the identification section in the PRISMA (figure 1), it was changed as: Records identified through database searching (PubMed, Scielo and Cochrane).
Section 3
Line 123 – beta-glucan and chitin are components of the cell wall not cell membrane – the same comment for lines 134
Lines 140 and 141 – TNF is not produced in C. glabrata
Answer: Thanks for your observation, the suggested changes have been made in lines 185 and 194 of the current paper. Additionally, the suggestions about TNF-alpha in lines 100 and 2021 were corrected.
Lines 151-152 – the authors have presented a set of genes and several proteins encoded by these genes but it should be written which gene is responsible for production of particular protein.
Lines 178-179 – this sentence (can obtain free iron or iron from proteins and promote resistance 178 to the death of macrophages using the siderophore-iron transporter Sit1) is not clear for me.
Answer: According to your comments, the wording has been corrected for better understanding in lines 237-241
“C. glabrata can acquire free iron and iron from iron-binding proteins such as hemoglobin, ferritin, and transferrin. The siderophore-iron transporter Sit1 is responsible for mediating the iron acquisition, giving the microorganism the ability to survive phagocytosis and replicate within the host’s macrophages due to the use of intracellular iron deposits”.
Lines 192 – 201 – this part is not clear for me (e.g., dependence between IL-1β production and CgYapsins should be explained).
Answer: According to your comments, the wording has been corrected for better understanding in lines 254-266.
“Rasheed et al. suggest in their study that CgYapsins (encoded by the CgYPS1-111 genes) inhibit the IL-1β production in macrophages so that the microorganism can proliferate and spread. The increased IL-1β output is part of the host’s immune system’s response to infection [89]. The study conducted in a murine model observed that a mutant strain of C. glabrata lacking the coding genes for CgYapsins showed less virulence and died inside the macrophages. Therefore, the authors suggest that given the absence of CgYapsins, the IL-1β-dependent inflammatory response is not inhibited inside the macrophages. Consequently, the microorganism dies as it lacks the aspartyl proteases that contribute to its survival”.
“The dependence between the IL-1β output and CgYapsins occurs because yapsins inhibit IL-1β production in macrophagues. Thus, the pathogen survives the host’s defense mechanism, proliferating and subsequently spreading. Conversely, in the absence of yapsins, there is no inhibition of IL-1β, and the pathogen dies inside the macrophages”
Section 4
Line 204 – could the authors give some examples of isolation of C. glabrata resistant to polyenes? Because of the mode of action resistance to this class of antibiotics is rarely observed.
Answer: Based on the comment made, lines 381-384 were addressed, adding the following reference:
Hull CM, Parker JE, Bader O, et al. Facultative Sterol Uptake in an Ergosterol-Deficient Clinical Isolate of Candida glabrata Harboring a Missense Mutation in ERG11 and Exhibiting Cross-Resistance to Azoles and Amphotericin B. Antimicrobial Agents and Chemotherapy 2012; 56: 4223–4232.
We have added the redaction of this observation as follows:
“Interestingly, Hull et al. 2012 found that the isolate of C. glabrata (CG156) has an ERG11 mutation that induces loss of function associated with cross-resistance to azoles and polyenes. This isolate exchanges ergosterol from the membrane for other sterols such as lanosterol and fecosterol, among others”
Line 216 “due to their low cost” – in my opinion there are many other important advantages (more important than low cost) of azoles, including low toxicity and good water solubility (fluconazole) – it must be mentioned in scientific paper.
(Bodey GP. Azole antifungal agents. Clinical Infectious Diseases; 14. Epub ahead of print 1992. DOI: 10.1093/clinids/14.Supplement_1. S161; Sheng C, Zhang W, Ji H, et al. Structure-Based Optimization of Azole Antifungal Agents by CoMFA, CoMSIA, and Molecular Docking. Epub ahead of print 2006. DOI: 10.1021/jm051211n. Sheehan DJ, Hitchcock CA, Sibley CM. Current and Emerging Azole Antifungal Agents. CLINICAL MICROBIOLOGY REVIEWS 1999; 12: 40–79.).
If authors are cited (e.g., lines 222 and 225) the year of publication also should be provided
Answer: We appreciate the observation made, we corrected the error in lines 287 and 290.
Lines 227 – 232 – but how many strains from Ghana were investigated in this study (strains were isolated from 176 pregnant patients – does it mean that there were 176 isolates?). I have some doubts about results concerning resistance to nystatin (but of course it is not a critical remark to the authors of this publication).
Answer: This is an excellent observation. Corrections were made on lines 292-301.
“Waikhom et al., 2020 analyzed in Ghana clinical isolates obtained from 176 pregnant patients. Only 54 patients were diagnosed with Candida infection with positive isolates (44 symptomatic and ten asymptomatic). C. glabrata was isolated in 25 symptomatic women and six asymptomatic women, being the most common isolation with 57.4%. Six C. glabrata isolates were susceptible to fluconazole (19.4%), 13 were susceptible-dose-dependent (41.9%), and 12 were resistant (38.7%). No C. glabrata isolate was susceptible to nystatin, 27 were susceptible-dose-dependent (87.1%) and four were resistant (12.9%). Seventeen strains were susceptible to voriconazole (54.8%), three susceptible dose-dependent strains (9.7%), and 11 strains were resistant (35.5%). [105]”
Line 312 – why the authors have presented this form of citations ((Vermitsky and Edlind, 2004).)
Answer: Thank you for this suggestion, the error was because it was not removed in the previous version of the document, because it served as a guide. Line 392.
The authors performed detailed analysis of publications concerning epidemiology, resistance and virulence of C. glabrata – from my study of literature I know that there are many other publications concerning investigation of C. glabrata isolated in Europe (I am mostly interested in this region) some of them should be cited. In some of these publications authors also investigated mechanisms responsible for resistance.
“Also, from a multihospital study, Aldejohann et al. reported resistence in the C. glabrata complex to several drugs including echinocandins, probably related to gene regulation; specifically, those linked to glucan synthase expression.[109] Similarly, Coste et al. observed in Switzerland resistance to echinocandins linked to the FKS-genes mutation.[110] However, other authors, such as Fraser et al., suggest that resistance is a rare phenomenon in some countries like the United Kingdom.[111] A multicenter study showed the relevance of C. glabrata in the yeast infections with mixed agents and suggested a potentiation of resistance in those cases.[112] In Jerusalem, Israel et al. examined C. glabrata strains that exhibited a susceptible-dose-dependent pattern to fluconazole with MIC values ≤ 32 μg / ml. [11]”
Short comment to information presented between lines 304-327 (overexpression of drug transporters) – interesting results in this area were presented by Gucwa and co-workers (2015 J Med Microbiol) who observed up regulation of CDR1 gene as a reason of azole resistance among C. glabrata strains isolated in Poland. Up regulation of CDR2 and ERG11 was not observed. Moreover, the authors did not observe any mutations within EGR11 gene – these mutations are identified as important mechanism of azole resistance among C. albicans.
Szweda et al. showed, by means of real-time PCR studies, that 13 of 15 azole-resistant strains had upregulation of the CDR1 gene encoding the efflux pump. While, no upregulation of expression of the CDR2 or ERG11 gene was observed. [125]
Line 352 - this sentence is not clear for me - “Pan drug resistance to these antifungals”
Answer: This is an excellent observation. The change was made on lines 434-435.
“Pan-resistance to these antifungals in C. glabrata has a prevalence range of 2 - 13%”
Lines 359 – 363 and 365 – 371. I would be grateful for a short comment. In both cases deletion of target genes resulted in increase of susceptibility. But is this the proof that proteins encoded by these genes are crucial for resistance?
Answer: This is an excellent observation, the brief comment was made between the lines 457-462,
“Accordingly, Roetzer et al., 2008 demonstrated that the protein VPH2, required for the vacuolar H+-ATPase function, was widely induced under different oxidative stress conditions in C. glabrata strains. These studies allow us to understand that, when subjected to certain conditions, the C. glabrata strains can induce the expression of this protein to maintain the internal pH of the cell and preserve its virulence”
Line 393 – gene’s delegation? (not clear), beside of that which gene?
Lines 397-400 – how many strains were investigated?
“Regarding C. bracarensis, Małek et al. performed in vitro susceptibility tests of different antifungals in isolates obtained by PCR from a group of 353 strains of the C. glabrata complex,”
Section 5
Line 427 – the title of Figure 2 should be translated (it is not in English)
Answer: Thanks for this observation, the translation error was fixed. Candida glabrata drug resistance and virulence mechanisms. Line 591.
Line 432 – it is relevant
Answer: Thanks for this observation. the mistake was corrected. Line 597.
Line 447 – why the cell wall is hydrophobic (it is composed of “carbohydrates”)
Answer: This is an excellent observation. The justification for why the cell wall is hydrophobic in Line 611, is explained in the next paragraphs
Además de Groot et al., 2020 demostraron que el incremento de la hidrofobicidad se veía asociada a la incorporación de diferentes adhesinas.
Section 6
Line 513 – do the authors mean any particular tissue? I would propose using plural in this place.
Line 547 – 548 – how many strains were investigated in these studies
Answer: Thank you for this suggestion, the number of strains in these studies were added. Lines 709-710.
Line 549 – why capital letter
Answer: Thanks for the correction, the requested change was made. Line 712.
Lines 551-560 – the percentage is not important if we do not know the number of isolates.
Answer: Thanks for the correction, the number of isolates that were used in the study was placed. Lines 715-719.
“Pereira et al. in a group of 50 healthy patients and another 50 patients with stomatitis and different types of lesions, showed that the isolates of C. glabrata from healthy patients (12 isolates) had moderate proteinase activity in 17%, and the remaining 83% showed no such activity; those from patients with stomatitis (24 isolates)”
Line 563 why capital letter (They??) – the same problem in line 564
Answer: Thank you for this observation, the error on line 727 was corrected.
Line 570-572 – again, it is important to know the number of strains tested
Answer: Thank you for this suggestion, The number of strains used in the studies were recorded. Line 731-735.
“however, Figueiredo-Carvalho et al. analyzed one strain of C. nivariensis, from a runny nose and observed elevated protease activity [168]. Finally, the most recent study conducted by Hernando-Ortiz et al. with strains of C. glabrata, C. nivariensis and C. bracarensis (two strains for each species)”
Line 599 – it is too general information
“Kalaiarasan et al. observed that 15/51 (29.41%) of their Candida spp. isolates showed phospholipase activity; however, only one isolate (4.3%) corresponded to C. glabrata.”
“A study conducted in Nepal with 71 isolates of Candida spp., showed only nine of them were C. glabrata (12.67%), and only three isolates presented phospholipase activity. In Turkey, from 50 isolates, 14 belonged to C. glabrata (28%), and only five strains showed phospholipase activity. Similarly, a study in Turkey with 100 isolates of Candida spp., showed only 9 C. glabrata isolates (9%) and three strains with phospholipase activity.”
Lines 634-639 – the authors must give the number of strains tested (percentage is useless if we do not know the number of isolates that were considered in these studies)
Answer: Thank you for this suggestion, The number of strains used in the studies were recorded. Line 796-808.
“There are studies that show that C. glabrata has poor hemolytic activity as well as inability to import heme efficiently compared with Candida albicans, so it is suggested that C. glabrata it depends mainly on the circulating sources of Fe in the host to be able to meet its needs of this micronutrient [190].”
“Similarly, Kalaiarasan et al. observed the production of hemolysins in 87% (20/23 strains) of the C. glabrata strains analyzed [145]. Vieira de Melo et al. also reported the production of hemolysins in all strains (7/70 strains) analyzed in their study and observed that the C. glabrata strains had a higher production than the C. albicans isolates and other species [171].”
Section 7
Lines 742-743 – how many strains were investigated
Answer: Thank you for this suggestion, The number of strains used in the studies were recorded. Line 913-915.
“On the other hand, studies such as those from Atalay et al. (9/50 strains) and Yagmur et al. (12/99 strains) do not report biofilm formation in any of the C. glabrata isolates analyzed [148, 176].”

Reviewer 2 Report
The Authors submitted a literature review regarding the antifungal resistance and virulence factors of Candida glabrata. The paper is well written and clearly illustrates the topic. Besides, the Authors reported adequate references to related works. Regarding the two sections "variation in susceptibility per Continent" and "variation in drug resistance due to the type of Candida and genetic variations", I suggest to the Authors to add two different tables reporting the main information. This allows the reader to have clearer and faster information to consult. After minor revision, the manuscript could be suitable for publication in the Journal. Please, find attached my comments.

Author Response
Answers to Reviewer 2 concerns:
The Authors submitted a literature review regarding the antifungal resistance and virulence factors of Candida glabrata. The paper is well written and clearly illustrates the topic. Besides, the Authors reported adequate references to related works. Regarding the two sections "variation in susceptibility per Continent" and "variation in drug resistance due to the type of Candida and genetic variations", I suggest to the Authors to add two different tables reporting the main information. This allows the reader to have clearer and faster information to consult. After minor revision, the manuscript could be suitable for publication in the Journal. Please, find attached my comments.
Answer: We appreciate the time and effort you have invested in the revision of our manuscript. Indeed, all your suggestions have improved the quality of our manuscript. In the main text, the additions are highlighted in yellow. We hope that we have correctly addressed all your concerns.
Answer: Added the two tables that you suggested in the sections Table 1. "Susceptibility variations of Candida glabrata complex per Continent", line 536 and Table 2. "Drug resistance fluctuations caused by the type of Candida and genetic variations", line 540.

Round 2
Reviewer 1 Report
The authors have answered all my comments. In my opinion the manuscript can be accepted in current form.